# Assessing the Impact of Work Activities on the Physiological Load in a Sample of Loggers in Sicily (Italy)

**DOI:** 10.3390/ijerph19137695

**Published:** 2022-06-23

**Authors:** Federica Masci, Giovanna Spatari, Sara Bortolotti, Concetto Mario Giorgianni, Laura Maria Antonangeli, John Rosecrance, Claudio Colosio

**Affiliations:** 1Department of Health Sciences, International Centre for Rural Health of the Santi Paolo e Carlo ASST of Milan, University of Milan, 20142 Milano, Italy; claudio.colosio@unimi.it; 2Department of Environmental and Radiological Health Sciences, College of Veterinary Medicine and Biomedical Sciences, Colorado State University, Fort Collins, CO 80523, USA; john.rosecrance@colostate.edu; 3Department of Biomedical, Dental and Morphological and Functional Imaging, University of Messina, 98122 Messina, Italy; gspatari@unime.it (G.S.); mariogiorgianni@virgilio.it (C.M.G.); 4University of Milan, 20122 Milano, Italy; bortolottisara96@gmail.com (S.B.); lantonangeli@unimi.it (L.M.A.)

**Keywords:** biomechanical overload, heart rate, loggers

## Abstract

Occupational logging activities expose workers to a wide range of risk factors, such as lifting heavy loads, prolonged, awkward positioning of the lower back, repetitive movements, and insufficient work pauses. Body posture has an important impact on the level of physiological load. The present study involved a group of 40 loggers in the province of Enna (Sicily, Italy) with the aim of defining the impact of logging activities on the workers’ physiological strain during the three primary work tasks of felling, delimbing, and bucking. The Zephyr Bioharness measurement system was used to record trunk posture and heart rate data during work tasks. The NASA TLX questionnaire was used to explore workers’ effort perception of the work tasks. Based on our results, the most demanding work task was tree felling, which requires a higher level of cardiac cost and longer periods spent in awkward trunk postures. The perceived physiological workload was consistently underestimated, especially by the more experienced loggers. Lastly, as the weight of the chainsaw increased, the cardiac load increased.

## 1. Introduction

Due to its high percentage of fatalities and serious injuries, the forestry sector is considered one of the most physically demanding and dangerous occupations across the world [1]. Based on the literature, forestry workers are primarily involved in chainsaw operations that are classified as heavy workload activities [2]. Logging generally includes three primary work tasks, which are designated as felling, delimbing, and bucking. Felling (Figure 1) consists of cutting down the tree; delimbing (Figure 2) refers to cutting or trimming branches, as well as the upper part of the tree, and bucking (Figure 3) consists of cutting the tree trunk into smaller pieces.

The logging activities expose workers to a wide range of risk factors, such as the lifting of heavy loads, prolonged adoption of awkward postures, repetitive movements, and a lack of breaks, which can cause musculoskeletal disorders (MSDs) [3]. According to European statistics, MSDs are responsible for long-term absences from work [4], and their incidence rate in Italy has systematically grown, especially in the forestry sector [5]. Specifically, our latest research study on a sample of Italian loggers indicated that tree delimbing and felling tasks were significantly associated with the risk of biomechanical overload in the low back [5]. Additionally, the literature indicates that high levels of physiological workload among chainsaw operators are associated with awkward body postures during the activity. Most notably, the literature indicates that the highest cardiovascular responses were associated with felling tasks that often require loggers to work in postures involving moderate to severe trunk and knee flexion [6]. Therefore, attention to body mechanics is fundamental to work design interventions aimed at reducing the physiological effort of professional loggers. 

Occupational research in the field requires innovative methodologies to assess anatomical and cardiovascular parameters, especially in challenging environmental work conditions. Heart rate is often used as an effective method of determining the physiological workload among workers in applied field situations [7]. Assessing direct measurements, such as heart rate, assists with the quantification of risk exposures. Currently, there are several technologies that are useful for collecting quantitative health-related data in field studies. Examples include wearable microsensors that are attached to the wrist or chest, which can record heart rate, breathing rate, and activity levels. Also commonly available is smart clothing, which embeds microsensors in clothing fabric to record electrocardiography signals, skin temperature, and body positions relative to gravity [8]. The use of microsensor technology (e.g., Zephyr Bioharness 3) was demonstrated to be useful in the assessment of trunk inclination during logging tasks [5]. The Bioharness 3 has also been used to assess heat-related illness [9,10], physiological status [11], and work-related stress [12].

Assessing the physiological workload entails the need to investigating operators’ subjective perception of fatigue in order to formulate the most suitable strategies to reduce the risk of biomechanical overload [13]. 

The aims of the present study were to: (a) investigate the impact of logging on physiological workload in a sample of chainsaw operators involved in tree felling, delimbing, and bucking activities; (b) explore the association of physiological workload with trunk posture and personal and occupational factors; and (c) explore the workers’ effort perception. 

## 2. Materials and Methods

A total of 65 workers were randomly chosen among the loggers active in the province of Enna and employed in the Department of Forestry of the Region of Sicilia. Study participants had to be between the ages of 18 and 60 and have at least three years of experience as a logger performing tree felling, delimbing, and bucking. Exclusion criteria included a history of cardiovascular or respiratory disease. The Forestry Department’s Health and Safety manager asked the subjects to participate in the study and sent them an official invitation via e-mail. Twenty-four workers declined the invitation, and one worker was dropped after enrollment because he could not complete the scheduled data collection shifts. The study took place during the second week of November in Sicily, at the Minerario of Guastella Park, Armerina Platz, and Altesina. Data collected from each participant included age, sex, work experience, history of participation in physical sports, secondary work if appropriate, the worker’s height and weight, and administration of the NASA Task Load Index questionnaire. During the subjects’ work activities, we recorded trunk postures and heart rate with the Zephyr Bioharness 3 instrumentation. Lastly, we also recorded specific characteristics of the chainsaws used, including model and weight. We defined the chainsaw as “light” or “heavy” based on a cutoff determined by their average weight. 

The study protocol was approved by the Ethical Committee of the University Hospital “G. Martino” of Messina. Prior to data collection, all subjects were informed of the purpose of the study and signed an informed consent declaration.

### 2.1. Zephyr Bioharness 3

The Zephyr Bio Module BH3 chest strap was worn by each worker (Figure 4) to assess the sagittal trunk posture and heart rate. At the beginning of the data collection (but prior to working), the resting erect trunk posture of each subject was recorded to establish a normalized baseline for all subjects. The normalized position was defined by having the worker stand in an anatomical position with their heels, buttocks, upper back, and head against a vertical wall for a period of 30 s. The trunk posture in that position was defined as 0 degrees of inclination. Forward inclination was in the positive direction and backward leaning was in the negative direction. The resting heart rate of each participant was recorded for 15 min while they were seated before the data collection started.

Trunk posture and HR data were collected for 30 min while each worker performed the three standardized logging work activities of felling trees, delimbing trees, and bucking (cutting sections of) logs. 

### 2.2. Qualitative Assessment: NASA Task Load Index and Borg Scale

The NASA TLX is a method commonly used to measure workers’ mental workload associated with cognitive tasks. It is a subjective, multidimensional questionnaire used to assess perceived workload on six dimensions, including mental demand, physical demand, temporal demand, performance, effort, and frustration [13]. For each logging task (felling, delimbing, and bucking), the worker was asked to assign a rating from zero to 20 (were zero = no demand and 20 = maximum demand) to the six aforementioned dimensions. 

According to the NASA TLX procedures, the results of this qualitative assessment were classified into four levels of workload, as follows:0–5: low;6–10: middle-low;11–15: middle-high;16–20: high.

### 2.3. Data Processing

The trunk postures and HR data recorded from the Zephyr Bioharness 3 were downloaded into an Excel spreadsheet for each worker, separated by each task. The avarage HR data during the work activities performed were obtained using the OMNISENSE™ 5.1 software (Medtronic, Minneapolis, MN, USA) [14]. 

#### 2.3.1. Heart Rate

From the HR data, we were able to estimate three parameters for determining physiological workload:

(1) The absolute (ACC) and relative cardiac cost (RCC) with the following formulas [15]:ACC = HR work − HR restwhere HR work is the average of the raw heart rate measured during the different tasks, and HR rest is the heart rate in a resting position.RCC = (ACC/CC max) × 100where ACC is the absolute cardiac cost and CCmax is the maximum cardiac cost calculated as HRmax – HR rest. HRmax is obtained from the formula 220-age.

The above results were interpreted based on the Frimat Scale [16] shown in Table 1 and Table 2.

(1).The percentage of relative heart rate (%RHR) was calculated with the following formula [17,18,19,20]:
RHR (%) = (HR work − HR rest)/(HRmax − HR rest) × 100% 
where HR work is the average of the raw heart rate measured during the work period, HR rest is the resting heart rate, and HRmax is the maximum heart rate. In this equation, the HRmax is given by 208 − 0.7 ˆ age [21](2).The percentage of cardiovascular load (CVL) was calculated based on the formula:
% CVL = 100 ((WHR − RHR)/HRmax (8 h))
where WHR is the average working heart rate, RHR is the resting heart rate, and HRmax (8 h) is the maximum acceptable HR for a work shift of 8 h, that is, 1/3 (220 − age) + RHR [22,23,24]

%CVL was used to evaluate cardiovascular load or aerobic strain: <30% = acceptable level, no action required; 30–59% = moderate level, peak loads should be reduced within a few weeks; 60–99% = high level, peak loads should be reduced within a few months; 100% = intolerably high level, peak loads should be reduced immediately, or work must be stopped.

#### 2.3.2. Trunk Postures

The trunk inclination data were classified according to ISO 11226 and DIN EN 1005-4 [25] into the following 4 categories:<0: awkward extension posture0–30: neutral posture30–60: non-neutral posture>60: awkward flexion posture

A MATLAB software (The MathWorks Inc., Natick, MA, USA) program was developed and used to process the trunk inclination data gathered by the Zephyr Bioharness 3 in order to obtain the percentage of time spent in awkward extension, neutral, non-neutral, or awkward flexion postures.

### 2.4. Statistical Analyses 

All data were analyzed using IBM SPSS Statistics version 27 (IBM, Armonk, NY, USA). The Kolmogorov–Smirnov test was used in the investigation of hypotheses about the normality of the parameter distribution (*p* > 0.05). A repeated measures ANOVA was used to explore the differences of the absolute cardiac cost, relative cardiac cost, the percentage of relative heart rate (%RHR), and the percentage of cardiovascular load (%CVL) among the tasks of felling, delimbing, and bucking. An ANOVA procedure was also used to investigate how trunk posture and other personal and occupational variables, such as the worker’s age and body mass index, the chainsaw’s weight, and the subject’s working experience, second job, and NASA results, would affect the physiological workload. In addition, this study applied linear regression models to assess the relationship between lower back posture data (as the predictor variable) and the physiological workload (as the outcome variable), considering the four parameters we evaluated (ACC and RCC, RHR, CVL).

Statistical significance was set with *p* ≤ 0.5. We also explored whether subjective perception of the required effort and the workload were comparable to the quantitative assessment.

## 3. Results

### 3.1. Study Population

A sample of forty loggers was selected from among the working population of the Forestry Department of the province of Enna (Sicily), Italy. For each subject, anthropometric measurements (height, weight, body mass index (BMI)) and personal data (age, working experience) were recorded, and are outlined in Table 3.

The mean age of the study population was 53.0 years (SD 4.5). The mean height and weight were 172.0 cm (SD 7.5) and 82.0 kg (SD 13.8), respectively. The mean BMI was 27.6: nobody was classified as underweight (BMI < 18.4), 14 workers were classified as normal weight (18.5 < BMI < 24.9), 16 were overweight (25 < BMI < 29.9), 9 fell into the category of light obesity (30.0 < BMI < 34.9), and 1 into the category of medium obesity (35.0 < BMI < 39.9). 

The sample of recruited loggers had a mean working experience of 27.0 years; based on this classification, 19 workers had less than 27.0 years of working experience, while 21 had more than 27.0 years of working experience. 

Thirteen (32.5%) workers did not have a secondary job, while 15 workers (37.50%) had a secondary job in the primary sector (agriculture), 9 in a secondary sector (mainly in the construction industry), and 3 workers in a tertiary sector (such as food or transport). 

The minimum chainsaw weight was 3.1 kg and the maximum 8.3 kg. The average weight of the chainsaws was approximately 7.3 kg, which we considered as the cutoff between light and heavy chainsaw equipment. Most workers (31) used a heavy chainsaw (>7.3 kg), while nine workers used a light chainsaw.

### 3.2. Heart Rate and Trunk Posture Data Derived from the Zephyr Bioharness 3 Instrumentation

The mean recorded HR for all subjects was 136.7 bpm during the felling task, 143.5 bpm during the delimbing task, and 143.5 bpm for the bucking task. Using the HR data, we obtained the average ACC, RCC, RHR, and CVL for each worker, as summarized in Table 4.

High levels of heart rate were found in all of the tasks performed, but slightly greater levels were registered during the delimbing task compared to felling and bucking. In particular, the ACC was 52.4, 55.4, and 51.5, respectively, for the felling, delimbing, and bucking tasks. According to the Frimat Scale [16], ACC values greater than 50 bpm are associated with severe activity. Similarly, the RCC values we obtained for the three tasks (respectively, 63.2, 67, and 61.5) identify an intense job. The %RHR was 61.2, 64.8, and 59, respectively, for felling, delimbing, and bucking. Since the %CVL values ranged between 38.3 and 40.3 for the three tasks, all working activities can be classified at a moderate level. There were no statistically significant differences in physiological workload parameters between the tasks (*p* > 0.05).

The linear regression analysis indicated a strong association between trunk postures and the physiological workload. In particular, we noticed (Table 5, Table 6, Table 7 and Table 8) that many of the physiological workload indicators were associated with neutral and non-neutral trunk inclination (0°–30° and 31°–60°, respectively) in the tasks analyzed. However, awkward postures were associated with higher ACC (*p* = 0.02), RCC (*p* = 0.02), %RHR (*p* = 0.02), and CVL (*p* = 0.04) only for the felling task. Despite these statistically significant strong associations, there was a limited amount of variability explained by the trunk postures. As shown in Table 5, Table 6, Table 7 and Table 8, the R^2^ values reach a greater percentage only for the trunk inclination ranges of 0°–30° and 31°–60°.

### 3.3. Personal, Occupational Factors and Cardiac Cost

The present study also investigated whether the workers’ personal characteristics (age, BMI, working experience, secondary job) and the chainsaw weight had an effect on the physiological workload in the different working tasks (felling, delimbing, and bucking). Younger loggers (<55 years) had a greater CCA, %RHR%, and CVL than older loggers, in particular during tree delimbing. Higher values of CCR were identified in older workers only for the tasks of felling and delimbing. 

We noticed that overweight (light obesity) workers had higher levels of ACC, RCC, %RHR, and CVL, compared with loggers of a normal weight. Interestingly, our results show a downward trend in the physiological workload in association with the level of work experience. Nevertheless, non-statistically significant differences were found between the physiological workload variables during the three tasks in association with age, BMI, working experience, and being involved in a secondary job (*p* > 0.05).

Heavy chainsaw weight was associated with higher levels of physiological workload. In particular, statistically significant differences (*p* < 0.05) were found for ACC, RCC, and RHR when workers used a lighter chainsaw (<7.3 Kg), compared to those using heavier chainsaws.

### 3.4. Effort Subjective Perception: NASA TLX Questionnaire

Comparing the six NASA TLX parameters, we found that mental demand had the highest score. Frustration seemed to have the lowest impact on the workload (Figure 5). We found that the physical demand was perceived to be higher among those reported to have a greater ACC during the delimbing task (*p* = 0.04). Interestingly, those reported to have an intense and severe ACC during bucking also assigned higher score to their performance, compared to those who reported a moderate ACC (*p* = 0.02, f = 2.73, Mean Square = 40.93, DF = 6). In parallel, the same workers with higher ACC and RCC also reported higher levels of physical demand (respectively *p* = 0.01, F = 3.39, Mean Square = 43.42, DF = 6 and *p* = 0.015, F = 3.31, MS = 44.8, DF = 5) for the same task. Lastly, those with moderate levels of CVL reported higher physical demand (*p* = 0.018, F = 4.45, MS = 66.2, DF = 2) and performance (*p* = 0.034, F = 3.7, MS = 61.7, DF = 2) than those reporting acceptable levels of CVL.

We also explored how chainsaw weight affected the NASA TLX load questionnaire response of the workers. For each of the NASA parameters, the sample population using the heavier chainsaw (>7, 2 kg) assigned middle-low and middle-high scores, compared to those using lighter chainsaw. However, non-statistically significant differences were found. Additionally, there were non-significant increases in mental demand for older workers (>55 years old). Our results showed an increase in physical demand, mental demand, and temporal demand in overweight workers, in comparison to workers of a normal weight. However, these differences were not statistically significant. As for the working experience, there was a tendency among experienced loggers to have lower scores for all the NASA items, compared to less experienced workers. A statistically significant difference was found for physical demand, where a higher percentage (19% vs. 0%) of workers with more experience reported a low physical demand (*p* = 0.04). Lastly, we explored how spending a higher percentage of one’s time in awkward postures may affect the NASA TLX load response of the workers, but only non-significant differences were found.

## 4. Discussion

The objectives of our study were to investigate the impact of the three main logging tasks on the physiological workload, and to explore its association with trunk posture and personal and occupational factors. Additionally, we assessed the integration of quantitative workload assessments with the subjective perceptions of loggers to identify solutions that reduce physiological and physical work stress. 

The average HR values measured in our study during felling, delimbing, and bucking were, respectively 136.7 bpm, 143.5 bpm, and 135.63 bpm. These HR results were slightly higher than those observed in Turkey by Grzywinski et al., who registered average values of heart rate during felling between 114.1 and 125.2 bpm among ten loggers using a chainsaw of 7.5 kg (average chainsaw weight in our study was 7.39 kg) [6]. Moreover, Grzywinski reported the highest cardiovascular response during the felling task, while our data showed higher values during the delimbing activities. A previous study also identified the delimbing task as the most strenuous and exhausting for loggers [26]. Higher cardiac costs for the delimbing task seem logical, as this task often requires the operator to stand in awkward postures for a prolonged period, while performing fast and repeated upper limb activity to cut the tree branches. 

As for the felling task (mean HR of 136.7 bpm), our results differ from those of another study conducted between July and August in the northeastern region of Turkey, where researchers reported a mean working heart rate of 122.8 bpm among ten chainsaw operators [27]. These differences may be explained by the seasonal variation in the data collection and the climate characteristics of the environment, as we conducted the study in November. Furthermore, investigating the time spent in awkward postures allowed the researchers to observe that a higher cardiac cost during the felling activities was correlated with a trunk inclination greater than 60°. These findings agree with previous studies that demonstrated the correlation between awkward posture and cardiac cost [6].

The %RHR of our workers ranged between 59.5% during the bucking task, and 64.8% during the delimbing task. These results are higher than the ones of Çalıkan and Çalar [27], which recorded the heart rate of ten loggers involved in forest felling operations throughout the working day. Their results showed the workers to have an average physical workload (%HRR) rate ranging between 39.5% and 50%. Our average %RHR for the felling task was 61.2% These differences can be explained by the different characteristics of the populations, such as age (53 years old vs. 39 years old in Calar et al.) and the BMI (ranging 18–39 in our population vs. 27–57), which are demonstrated to be positively associated with physiological workload. In addition, our study did not include risk assessment data for the entire shift and, according to Apud, Bostrand, Mobbs et al., %RHR at work should not exceed 40% for an 8 h period to avoid fatigue [28].

Our results showed that less experienced workers (<27 years) tended to have higher heart rates than the experienced workers. Although the difference was not statistically significant, we can suppose that the longer experience gained might lead the worker to self-improve their logging techniques, and therefore reduce cardiovascular strain. 

A previous study in agriculture [29] also compared HR and blood pressure before and just after the completion of work in a sample of farmers, to assess how physiological stress was related to age and BMI. There were no specific studies on the logging population available in the literature. Our study sample included loggers of normal weight, overweight loggers, and few cases of obese workers, but we did not find any statistically significant difference in HR associated with BMI. 

Our data also revealed that cardiovascular strain increases when a heavier chainsaw (>7, 3 kg) is used during felling and delimbing tasks. These results were consistent with Parker and Sullman [26], who compared the cardiovascular workloads of forestry workers in New Zealand. They reported higher heart rates among workers using large capacity chainsaws (9.5 kg) as compared to those using small and medium chainsaws (7.7 kg and 8.2 kg, respectively) during the deliming task. Eleven males and one female from the logging industry were equipped with a Sport Tester PE3000 heart rate monitor and results showed them to have higher HR values during delimbing. Therefore, we can state that using heavy chainsaws in logging activities can increase the cardiac cost. 

Worker fatigue perception was investigated through the NASA TLX questionnaire. Based on those results, we found that physiological workload resulted in an underestimation in fatigue perception among the 40 loggers. We noticed that those who had greater experience in the logging tasks perceived a lower physical demand. We can suppose that the greater experience gained by the workers might affect their fatigue perception.

Moreover, we found a tendency among overweight workers to declare a middle-high mental demand. These results are in line with the latest evidence in the literature [30] that highlighted that healthy lifestyle and diet, in addition to regular physical activity, can have a positive effect on effort perception and cardiovascular diseases, specifically. A study showed that poor physical fitness is associated with an impairment of cardiac vagal function, and that differences in heart rates can probably be attributed to fitness rather than age [31].

There are no specific studies examining the correlation between HR and risk perception of forestry workers, but a few studies in the literature have investigated general hazard perception. A study by Pecyna and Buczaj [32] suggested that, among a group of 135 forestry workers, the most relevant risk perception was hot and cold microclimates, biological risk, and contact with wild animals, followed by dangerous machinery, chemical substances, and awkward postures.

Given the results of this study, it seems crucial to adopt preventive strategies to reduce risk, including the implementation of effective training courses and the appropriate selection of lighter chainsaws. It has already been demonstrated in the literature [33] that safety courses are successful in preventing injuries and several diseases, as well as in increasing workers’ responsibility and awareness in the workplace.

## 5. Limitations

This study focused on assessing the physiological workload of loggers using only the heart rate index. Monitoring the breathing rate and body temperature would have provided researchers with more physiologic details to reinforce their conclusions. Our sample size of loggers was comparable with previous forestry sector studies. However, it is important to consider the small sample size in relation to the implications of our results for other logging populations that vary in age, BMI, experience, and physical fitness. Moreover, the low level of education of the loggers might have created some challenges in their understanding of the NASA questionnaire. In addition, we did not administer this questionnaire for each single subtask, but with reference to the whole shift. Lastly, individual logging techniques could have affected a logger’s heart rate, regardless of the specific task performed. 

## 6. Conclusions

Forestry workers are involved in physically and mentally demanding activities that can lead to an increased cardiac cost and, consequently, be responsible for higher physiological workload. This risk was also shown to be higher in association with the chainsaw’s weight (>7.3 kg). Conversely, effort perception resulted to to be underestimated by the forestry workers.

It is necessary therefore to focus on preventive solutions that consist of:(a)Provide the workers with ergonomic tools that are specific to the task being performed. For example, chainsaw size and weight should be specific to the logging tasks. The smallest and lightest chainsaws are recommended for the delimbing task, which involves removing small tree branches. The larger (heavier) chainsaws should be reserved only for tree felling, which involves cutting large diameter tree trunks.(b)Because felling tasks involve postures with trunk flexion greater than 60°, ergonomic modifications aimed at reducing awkward postures, as well as the provision of mechanized felling equipment, should be considered as part of the overall safety process.(c)If logging tasks are performed throughout the work shift, the physiological workload of workers should be monitored to prevent cardiovascular overload.

## Figures and Tables

**Figure 1 ijerph-19-07695-f001:**
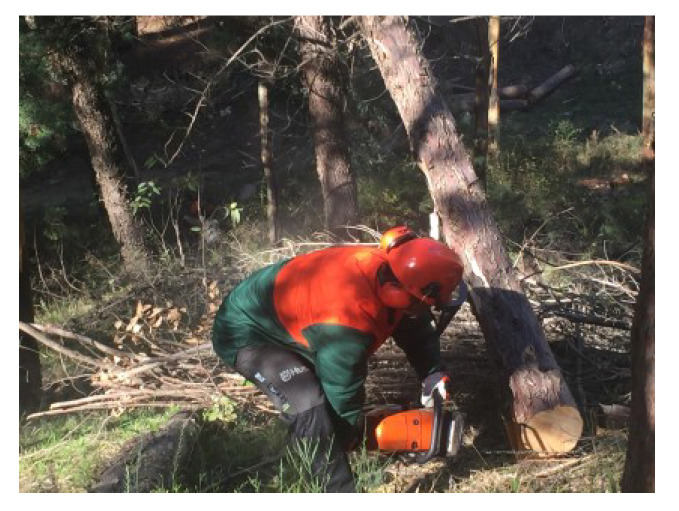
Tree felling task.

**Figure 2 ijerph-19-07695-f002:**
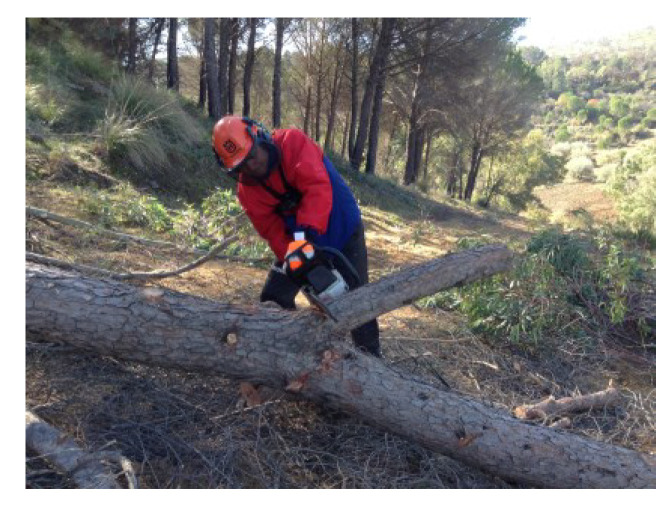
Tree delimbing task.

**Figure 3 ijerph-19-07695-f003:**
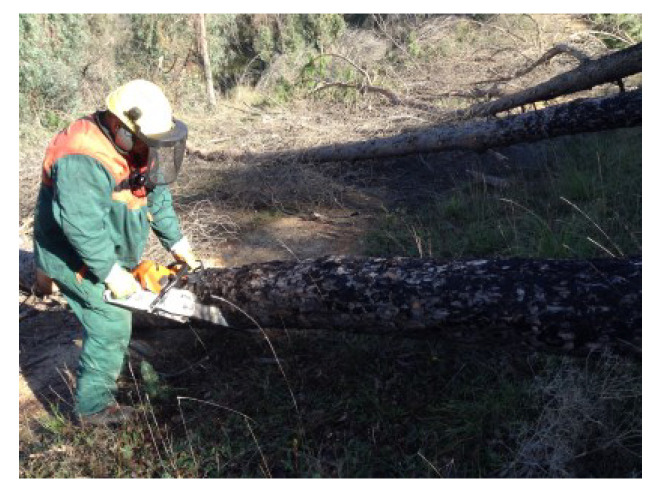
Tree bucking task.

**Figure 4 ijerph-19-07695-f004:**
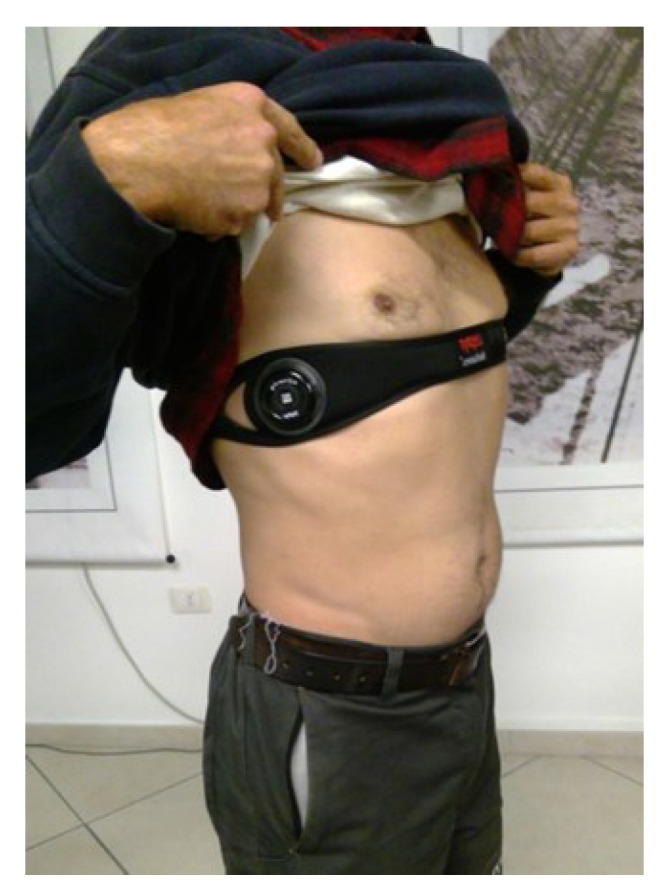
Zephyr Bioharness device worn by a worker.

**Figure 5 ijerph-19-07695-f005:**
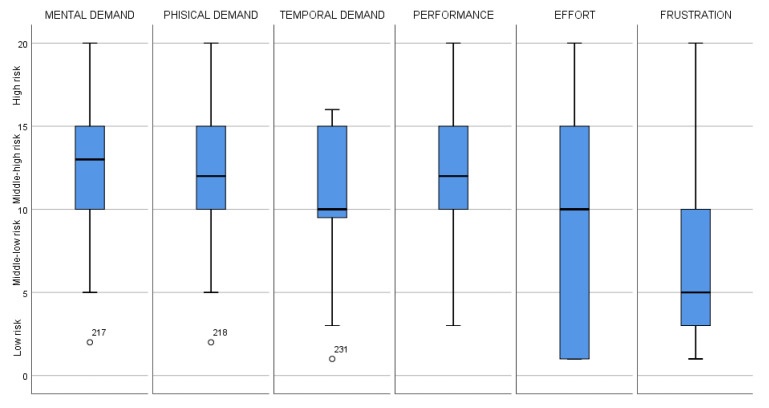
Comparison of the six NASA TLX parameters.

**Table 1 ijerph-19-07695-t001:** Absolute Cardiac cost (ACC) values for a job.

*Absolute Cardiac Cost (ACC) Values*	*ACC For a Job*
*0–9*	Very light
*10–19*	Light
*20–29*	Moderate
*30–39*	Heavy
*40–49*	Very heavy

**Table 2 ijerph-19-07695-t002:** Relative Cardiac cost RACC) values for a job.

*Relative Cardiac Cost (RCC) Values*	*RCC for a Job*
*0–9*	Very light
*10–19*	Light
*20–29*	Moderate
*30–39*	Heavy
*40–49*	A little heavy
*50–59*	Very heavy
*60–69*	Intense

**Table 3 ijerph-19-07695-t003:** Demographic information from the 40 loggers participating in the study.

	Min	Max	Mean	SD
Height (cm)	152.0	187.0	172.0	7.5
Weight (kg)	60.0	110.0	81.9	13.8
Age (years)	47.0	63.0	52.6	4.5
Body Mass Index	21.2	35.9	27.6	3.7
Working experience (years)	6.0	41.0	27.0	6.0

**Table 4 ijerph-19-07695-t004:** Comparison, across 40 loggers of absolute cardiac cost, relative cardiac cost, and relative heart rate, for felling, delimbing, and bucking tasks.

Physiological Workload Parameters	Felling			Delimbing			Bucking			*p*	DF	F
	Min	Max	Mean (SD)	Min	Max	Mean (SD)	Min	Max	Mean (SD)			
ACC	13.0	87.0	52.5 (22.2)	4.0	109.0	55.4 (24.7)	28.0	94.0	51.5 (24.7)	*p* ≥ 0.05	2	0.67
RCC	25.0	94.0	63.2 (28.0)	5.0	96.0	67.0 (28.3)	53.0	100.0	61.5 (30.6)	*p* ≥ 0.05	2	081
%RHR	24.0	89.6	61.2 (27.2)	5.0	110	64.8 (27.5)	51.7	97.1	59.5 (29.7)	*p* ≥ 0.05	2	0.84
%CVL	7.3	71.3	38.3 (17.1)	3.0	89.3	40.3 (19.3)	15.8	72.1	37.7 (18.7)	*p* ≥ 0.05	2	0.58

**Table 5 ijerph-19-07695-t005:** Linear regression model: trunk inclination classifications as predictive variables associated with ACC in the total sample (*n* = 40) for the three tasks of felling, delimbing, and bucking.

	Felling					Delimbing					Bucking				
Trunk Inclination	B	β	CI95%	*p*	R^2^	B	β	CI95%	*p*	R^2^	B	β	CI95%	*p*	R^2^
0°–30°	0.396	0.253	−0.031–0.824	0.06	0.089	0.622	0.458	0.249–0.995	0.01	0.242	0.436	0.284	0.23–0.85	0.03	0.113
31°–60°	1.276	0.459	0.531–2.02	0.01	0.251	0.760	0.298	−0.046–1.56	0.06	0.092	1.45	0.545	0.59–2.32	0.01	0.244
>60°	0.703	0.315	0.097–1.31	0.02	0.133	0.844	0.141	−1.03–2.71	0.36	0.023	0.115	0.434	−0.765–0.995	0.79	0.002

B = unstandardized regression coefficients; β = standardized regression coefficients; CI95% = confidence interval at 95%.

**Table 6 ijerph-19-07695-t006:** Linear regression model: trunk inclination classifications as predictive variables associated with RCC in the total sample (*n* = 40) for the three tasks of felling, delimbing, and bucking.

	Felling					Delimbing					Bucking				
Trunk Inclination	B	β	CI95%	*p*	R^2^	B	β	CI95%	*p*	R^2^	B	β	CI95%	*p*	R^2^
0°–30°	0.674	0.342	0.121–1.228	0.01	0.145	0.702	0.452	0.307–1.097	0.01	0.265	0.700	0.244	0.206–1.194	0.01	0.186
31°–60°	1.230	0.351	0.265–2.195	0.01	0.156	1.194	0.409	0.34–2.048	0.01	0.183	1.71	0.512	0.672–2.74	0.01	0.237
>60°	0.972	0.345	0.187–1.758	0.01	0.149	0.775	0.114	−1.209–2.76	0.43	0.017	0.206	0.062	−0.84–1.25	0.69	0.004

B = unstandardized regression coefficients; β = standardized regression coefficients; CI95% = confidence interval at 95%.

**Table 7 ijerph-19-07695-t007:** Linear regression model: trunk inclination classifications as predictive variables associated with %RHR in the total sample (*n* = 40) for the three tasks of felling, delimbing, and bucking.

	Felling					Delimbing					Bucking				
Trunk Inclination	B	β	CI95%	*p*	R^2^	B	β	CI95%	*p*	R^2^	B	β	CI95%	*p*	R^2^
0°–30°	0.652	0.340	0.113–1.19	0.01	0.143	0.682	0.190	0.297–1.06	0.01	0.263	0.676	0.365	0.195–1.156	0.01	0.184
31°–60°	1.190	0.349	0.250–2.131	0.01	0.155	1.150	0.411	0.316–1.948	0.01	0.179	1.65	0.514	0.645–2.66	0.01	0.235
>60°	0.933	0.341	0.168–1.699	0.01	0.145	0.727	0.956	−1.21–2.66	0.45	0.016	0.208	0.065	−0.816–1.323	0.68	0.005

B = unstandardized regression coefficients; β = standardized regression coefficients; CI95% = confidence interval at 95%.

**Table 8 ijerph-19-07695-t008:** Linear regression model: trunk inclination classifications as predictive variables associated with %CVL in the total sample (*n* = 40) for the three tasks of felling, delimbing, and bucking.

	Felling					Delimbing					Bucking				
Trunk Inclination	B	β	CI95%	*p*	R^2^	B	β	CI95%	*p*	R^2^	B	β	CI95%	*p*	R^2^
0°–30°	0.240	0.200	−0.95–0.576	0.15	0.055	0.466	0.439	0.161–0.771	0.01	0.201	0.281	0.242	−0.44–0.605	0.08	0.079
31°–60°	0.999	0.496	0.414–1.58	0.01	0.251	0.471	0.236	−0.189–1.13	0.15	0.055	1.07	0.533	0.396–1.75	0.01	0.222
>60°	0.492	0.287	0.015–0.968	0.04	0.108	0.769	0.165	−0.765–2.303	0.31	0.028	0.76	0.038	−0.615–0.767	0.82	0.001

B = unstandardized regression coefficients; β = standardized regression coefficients; CI95% = confidence interval at 95%.

## Data Availability

Data are available upon request to the authors.

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
