# Peer review of "Assessing the Impact of Work Activities on the Physiological Load in a Sample of Loggers in Sicily (Italy)"

_ijerph, 2022, doi:10.3390/ijerph19137695_

Round 1
Reviewer 1 Report
The research is interesting, and it is worth publishing after major revision to remove the limitations found in the applied Methods that will bring the research updated with respect to the current literature trends.
The declared paper's aims are:
“The aims of the present study were to: a) investigate the impact of logging on physiological workload in a sample of chainsaw operators involved in tree felling, delimbing and bucking activities; b) explore the association of physiological workload with trunk posture and personal and occupational factors; and c) explore the workers’ effort perception.”
As declared in the aims, the presented research introduces some interesting innovations introducing “the association of physiological workload with trunk posture and personal and occupational factors.”
Unfortunately, some limitations in the adopted methodology restrict the reliability of the obtained result.
Indeed, the method used to assess a worker’s workload through the estimate of the absolute cardiac cost (ACC) using the formula ACC = HR average – HR rest appears to be very basic, neglecting the age effect on HR and the subsequent effort level (ACC) classification on the base of average HR is referred to quite old publication Frimat et al. 1989.
For such a reason, the paper needs major revision to introduce a more accurate ACC assessment and more recent classification scales and compare and discuss the new computed results with those already presented in the paper.
Some suggestions for the more recently introduced average HR computation from the literature are given in the following.
Heart rate is measured to estimate an acceptable workload in physically demanding occupations. Average HR is a good indicator of physical stress and workload used in literature by many authors (Anwer et al., 2021).
Because heart rate is also inherently associated with age and resting heart rate, relative heart rate is the appropriate indicator for predicting the task demands that substantially reflect the subject’s age and resting heart rate.
The relative heart rate is positively associated with job demands, and therefore, it is the indicator of job demands at the individual level. Relative heart rate (RHR) is calculated based on the following equation defined by researchers in work physiology, ergonomics, and sports sciences (Kirk and Sullman, 2001; Wu and Wang, 2002) Rodahl, 1989)
??? (%)= (??????− ??????)/(?????−??????) ×100%
Where ?????? is the average of the raw heart rate measured during the work period, ?????? is the resting heart rate, ????? is the maximum heart rate.
It has been suggested that such ????? should be determined individually with specific measurement tests before the workload assessment. When such measurement is not available, the most credited ????? assessment is referenced by the equation introduced by (Tanaka et al., 2001).
?????(???)=207 – 0.7 × ???
Please introduce such RHR(%) assessment in the method and compare it with the ACC you assessed.
Regarding the classification of the workload effort (Adi and Ratnawinanda, 2017) classified fatigue levels according to the percentage of cardiovascular load (defined as CVL (%) = (HRwork-HRrest) / (HRmax-HRrest) x 100, i.e. the same as above) and recommended workers based on their CVL values. Notably, they classified workers with CVL values less than 30% as having no fatigue, while workers with CVL values between 30 and 60% were recommended to have rest breaks. Workers with CVL values between 60 and 80% and between 80 and 100% are supposed to have a short work period and special treatment, respectively. Those with CVL values greater than 100% should completely stop working.
Please introduce such CVL(%) assessment in the method and compare it with respect to the FRIMAT scale you used.
However, the reliability of the average HR parameter used as a surrogate to measure physical fatigue represents an open debate in the literature. Indeed recently, research showed that combining HR with other physiological measures could improve the prediction of fatigue. For example, (Umer et al., 2020) predicted 95% of physical fatigue levels using a combination of HR, thermoregulatory, and respiratory metrics in university students during a simulated construction task. However, the accuracy dropped to 57% if only HR data was used to predict fatigue. Similarly, (Aryal et al., 2017) reported a 72% prediction accuracy in estimating physical fatigue using combined findings of HR and skin temperature; however, the accuracy dropped to 59% if only HR data was used. These results highlight the benefit of using combined metrics to predict physical fatigue in individuals.
In addition to HR, HRV is a metric of beat-to-beat variation of HR and is found to be a strong marker of cardiac health. The measurement of HRV may be an important metric to measure physical fatigue because a diminished high frequency component of HRV value may indicate heavy physical loads or fatigue in construction workers (Anwer et al., 2021).
In the discussion section, please introduce why such a multiple simultaneous measurement approach has not been used in the research even if this could be approached with the Zephir device.
Please discuss the eventual differences or confirmations you will highlight comparing the new and previous acc assessment and related classification.
Adi, T. J., Wahyu, and Ratnawinanda, A., Lila (2017). Construction Worker Fatigue Prediction Model Based on System Dynamic. MATEC Web Conf. 138, 05004. doi: 10.1051/matecconf/201713805004.
Anwer, S., Li, H., Antwi-Afari, M. F., Umer, W., and Wong, A. Y. L. (2021). Evaluation of Physiological Metrics as Real-Time Measurement of Physical Fatigue in Construction Workers: State-of-the-Art Review. J. Constr. Eng. Manage. 147, 03121001. doi: 10.1061/(ASCE)CO.1943-7862.0002038.
Aryal, A., Ghahramani, A., and Becerik-Gerber, B. (2017). Monitoring fatigue in construction workers using physiological measurements. Automation in Construction 82, 154–165. doi: 10.1016/j.autcon.2017.03.003.
Kirk, P. M., and Sullman, M. J. (2001). Heart rate strain in cable hauler choker setters in New Zealand logging operations. Appl Ergon 32, 389–398. doi: 10.1016/s0003-6870(01)00003-5.
Tanaka, H., Monahan, K. D., and Seals, D. R. (2001). Age-predicted maximal heart rate revisited. Journal of the American College of Cardiology 37, 153–156. doi: 10.1016/S0735-1097(00)01054-8.
Umer, W., Li, H., Yantao, Y., Antwi-Afari, M. F., Anwer, S., and Luo, X. (2020). Physical exertion modeling for construction tasks using combined cardiorespiratory and thermoregulatory measures. Automation in Construction 112. doi: 10.1016/j.autcon.2020.103079.
Wu, H.-C., and Wang, M.-J. J. (2002). Relationship between maximum acceptable work time and physical workload. Ergonomics 45, 280–289. doi: 10.1080/00140130210123499.
Rodahl, K. (1989). The Physiology of Work, London, Taylor & Francis.
Reviewer 2 Report
The manuscript deals with the interesting topic related to assessing the impact of logging activities on the physiological workload in a sample of chainsaw workers. The paper requires a major revision and corrections to improve the overall structure and readability (extensive refinement in results and conclusions). I have following questions and suggestions for the authors:
L2-3: the authors are suggested to rephrase the title of the paper according to the case study approach as the sample does not contain a representative number of workers either at the level of Sicily or at the level of Italy!
L31: I suggest the authors to check the citation (5) in the subject sentence. If the source used is good, the numbering statement should be corrected throughout the paper!
In introduction chapter, did I miss disposition paragraph?
In methods chapter sample method used and the accompanying explanation is not clearly written! Please include the same and clarify better!
L94: in chapter 2.1 I ask the authors to clarify the calibration of the device in question, i.e. the values obtained by measurement with respect to the anthropometric values of the sampled subjects.
L140: In the statistical analyses it is stated that a repeated measures ANOVA was used, but in the results the same is not visible (only p-value is shown, where is F-value, df, values related to post-hoc test?). Also, a test to examine the normality of the distribution was not performed!? The explanation for the regression (one dependent and one independent variable) is not given in the subject chapter, and the same was used in Figures 7 to 10!? I ask the authors to include the above in their work!
L175: in resalts chapter (in text and in figure title) write the correct name of the abbreviation for Absolute Cardiac Cost (ACC)!
L177 and L179: duplication of data in Table 2 and Figure 6! I ask the authors to decide on one presentation! if it is a table, then specify the min and max value. If it is a figure, then specify the names of the x and y axes and the legend for the values shown in the figure!
Figure 7 to 10: R2 is very small, particularly insignificant in Figure 8. It is suggested that the authors present the obtained values in a table, and that Figure 8 should be omitted from the presentation of the results! If the images remain in the article, they should correct the duplication of the name of the y axis and correct the abbreviation for ACC in the title of the figure!
L186-L193; L216-219; L224-L227: which test is used here? In the results, in addition to the p-value when conducting the test, the type of test that was applied should be stated, as well as other values that are relevant for the same!? I ask the authors to correct the same through the presentation of all results!
L245: The discussion needs to be supported by additional sources i.e. research conducted in Western European countries and beyond.
L266-267: Is the statement within the sentence in question well written since R2 was extremely small in figure 8!? I ask the authors to rephrase the above!
L324: Conclusion chapter - general potential guidelines for improvement are given here, and no specific conclusions emerging from the research results. I would like to see more precise conclusions, evidently matching your purpose. Please rephrase the conclusions!
Author Response
the manuscript deals with the interesting topic related to assessing the impact of logging activities on the physiological workload in a sample of chainsaw workers. The paper requires a major revision and corrections to improve the overall structure and readability (extensive refinement in results and conclusions). I have following questions and suggestions for the authors:
L2-3: the authors are suggested to rephrase the title of the paper according to the case study approach as the sample does not contain a representative number of workers either at the level of Sicily or at the level of Italy!
L31: I suggest the authors to check the citation (5) in the subject sentence. If the source used is good, the numbering statement should be corrected throughout the paper!
A: The citation used is correct, and the numbering statement as well. We added the missing information of the journal. We deleted the reference in the materials and methods.
In introduction chapter, did I miss disposition paragraph?
A: The purpose of the study (study aims) is presented in the introduction (page 3, l 78-81). I report here the paragraph:
“The aims of the present study were to: a) investigate the impact of logging on physio-logical workload in a sample of chainsaw operators involved in tree felling, delimbing and bucking activities; b) explore the association of physiological workload with trunk posture and personal and occupational factors; and c) explore the workers’ effort perception.”
In methods chapter sample method used and the accompanying explanation is not clearly written! Please include the same and clarify better!
A: Thank you for the comment. The sample method has been clarified. The paragraph has been rewritten as follows:
Previous version:
“Forty forestry workers from the province of Enna, Sicily were recruited in the study. Inclusion criteria for participating in the study included being 18 to 60 years old and at least three years of working experience as a logger performing tree felling, delimbing and bucking. Exclusion criteria included a history of cardiovascular and or respiratory diseases.”
Current version:
“A total of 65 workers were randomly chosen among the loggers active in the province of Enna and employed in the Department of Forestry of the Region of Sicilia. Study participants had to be between the ages of 18 and 60 and have at least three years of experience as a logger performing tree felling, delimbing, and bucking. Exclusion criteria included a history of cardiovascular or respiratory disease. The Forestry Department's Health and Safety manager asked the subjects to participate in the study and sent them an official invitation via e-mail. Twenty-four workers declined the invitation, and one worker was dropped after enrollment because he couldn't complete the scheduled data collection shifts.”
L94: in chapter 2.1 I ask the authors to clarify the calibration of the device in question, i.e. the values obtained by measurement with respect to the anthropometric values of the sampled subjects.
A: None of the above measurements is correlated to the workers’ anthropometric data (high and weight) we collected.
The calibration of the device and the measurements principles are not related to the subject anthropometric data. The trunk posture data is obtained with a triaxial accelerometer imbedded in the Bioharness sensor. Trunk posture is derived from the accelerometer relative to gravity. The calibration procedures for the trunk posture measurements are explained in the method section. Heart rate data is obtained in the same manner as an electrocardiogram and is not influenced by anthropometric variables.
L140: In the statistical analyses it is stated that a repeated measures ANOVA was used, but in the results the same is not visible (only p-value is shown, where is F-value, df, values related to post-hoc test? Also, a test to examine the normality of the distribution was not performed!? The explanation for the regression (one dependent and one independent variable) is not given in the subject chapter, and the same was used in Figures 7 to 10!? I ask the authors to include the above in their work!
A: F-value, df, values related to post-hoc test were added to the paragraph of the results. We rewrote the material and methods sub session “statistical analysis” in order to describe the missing information ( page 6).
L175: in resalts chapter (in text and in figure title) write the correct name of the abbreviation for Absolute Cardiac Cost (ACC)!
A: Thank you for finding that error. We reviewed the manuscript to correct all the errors related to acronyms.
L177 and L179: duplication of data in Table 2 and Figure 6! I ask the authors to decide on one presentation! if it is a table, then specify the min and max value. If it is a figure, then specify the names of the x and y axes and the legend for the values shown in the figure!
A: In table 2 we added min and max values. In addition, in the same table we included data obtained from other physiological risk assessment methods as requested by Reviewer 1. Figure 6 was deleted.
Figure 7 to 10: R2 is very small, particularly insignificant in Figure 8. It is suggested that the authors present the obtained values in a table, and that Figure 8 should be omitted from the presentation of the results! If the images remain in the article, they should correct the duplication of the name of the y axis and correct the abbreviation for ACC in the title of the figure!
A: Figures 7 to 10 have been replaced with tables 5 to 8.
L186-L193; L216-219; L224-L227: which test is used here? In the results, in addition to the p-value when conducting the test, the type of test that was applied should be stated, as well as other values that are relevant for the same!? I ask the authors to correct the same through the presentation of all results!
A: Thank you for the comment and the suggestions. We added specific information about the test in the materials and methods subsection “Statistical Analyses”.
L245: The discussion needs to be supported by additional sources i.e. research conducted in Western European countries and beyond.
A: The discussion includes several references to similar logging studies from New Zealand, Turkey and India, Africa. We chose to only include studies which were directly related to our study aims. In all the respect to Reviewer 2, we feel our discussion reflects the studies most pertinent to the scope of our study and do not see a need of additional support.
L266-267: Is the statement within the sentence in question well written since R2 was extremely small in figure 8!? I ask the authors to rephrase the above!
A: Figure 8 has been deleted and replaced with a table…In the result
L324: Conclusion chapter - general potential guidelines for improvement are given here, and no specific conclusions emerging from the research results. I would like to see more precise conclusions, evidently matching your purpose. Please rephrase the conclusions!
A: Thank you for the valuable comment and suggestions regarding the conclusion section. As suggested, we rewrote the entire conclusion section to reflect the primary purpose and results of the study. Please see the new conclusion section.

Round 2
Reviewer 1 Report
English language and style are fine, but some minor spell check would improve the manuscript quality.
Reviewer 2 Report
The manuscript has been improved in accordance with my comments!